# Exploration of Influence of Socioeconomic Determinants on Mortality in the European Union

**DOI:** 10.3390/ijerph17134699

**Published:** 2020-06-30

**Authors:** Beata Gavurova, Samer Khouri, Viliam Kovac, Michaela Ferkova

**Affiliations:** 1Faculty of Mining, Ecology, Process Control and Geotechnologies, Technical University of Košice, 04001 Košice, Slovakia; samer.khouri@tuke.sk; 2Faculty of Economics, Technical University of Košice, 04001 Košice, Slovakia; viliam.kovac@tuke.sk (V.K.); ferkovamichaela@outlook.com (M.F.)

**Keywords:** standardised mortality rate, healthcare, healthcare system, gross domestic product, healthcare expenditure, unemployment rate, regression analysis, fixed effects regression model, random effects regression model, European Union

## Abstract

Economic performance measured through the gross domestic product indicator and the poverty rate varies across the whole European Union, together with the considerable income inequalities in the long-term. Economic growth may not bring a reduction in the health inequalities in the individual countries themselves. In order to eliminate health inequalities, the different types of policies implemented in the health, social and economic systems need to be explored in more detail. Mortality is explored through an indicator of the standardised mortality rate for both sexes explained by the several socioeconomic determinants, among which variables such as the variations of the gross domestic product per capita, the healthcare expenditures, the unemployment rate, and the healthcare system financing. Almost in all the described cases, these dimensions have negative impact. All the influences are expressed in a relative way in order to be suitably interpretable. The analysis is not comprehensive; nevertheless, it contains 18 regression models to cover as many aspects as it is possible. The Discussion section offers an evaluation of the obtained results according to the outcome of the other studies.

## 1. Introduction

Health inequalities have been a research issue for decades along with the significant differences between the countries in infant mortality, premature mortality, avoidable mortality, and the other health indicators. The poor living and working conditions, smoking, alcohol consumption, unhealthy food, and meal regime are more typical signs for the countries with shorter life expectancy. A social gradient has also been identified among the European countries. The population with lower education, employed in easier working positions, and earning lower income tend to die younger, and these inhabitants are more likely to experience serious health problems. This can be also applied to some ethnic minorities. Infant mortality is also higher in the lowest socioeconomic groups, and the inequalities in infant mortality have increased in many countries in the recent period. These consistent facts highlight the importance of studying the impact of the socioeconomic determinants on mortality of the population. The different countries pay attention to this issue, which is influenced not only by the economic development of the particular countries but also by the types of policies, healthcare systems, health insurance systems, and so forth. Looking at this issue in general, responsibility for addressing healthcare problems has usually been given to the health sector. It has been often associated with healthcare availability, the availability of resources, and the health behaviour of the population. It should be noted that the health status of the population is being altered often, and it is also dependent on the social programmes and the policies along with the economic behaviour of the country, and so forth. The socioeconomic determinants of health of the population are affected not only by government, civil society, and local communities but also by the business community and the international organisations. These institutions can help shape the active policies and the existing programmes that should synergistically cover all the key sectors of society, including the sectors related to the healthcare and social system. The health inequalities arise not only as the result of the social gradient but also by an uneven distribution of income, authority, goods and services that also create the disparities and the discrepancies related to access to healthcare, education, working and living conditions, and so forth. The improper health damaging environment is the result of incorrect social policies and programmes and policy decisions. The differences in the allocation of power and economic order at the global level possess great importance for equality in the health status of the population of the countries. Employment is also an important socioeconomic determinant, which has the effect of increasing or decreasing social disparities, as well as influencing national productivity. The main objectives of social and economic policies at both national and international levels are to ensure decent working conditions, fair employment, and full employment in the country. Safe and well-paid work, job opportunities and the optimal relationship between work and out-of-work activities have a significant impact on eliminating health inequalities. Exploring financial security, social status, personal development, and the development of social relations stand at the forefront of this field. A special determinant of health is the healthcare system, which is an influential and dependent variable in relation to other determinants of health status. However, responsibility for health of the population and health equality measures should be shifted to the governmental level and linked to all the active policies of the country. The health disparities should be one of the main indicators not only of the health system but also of the whole country performance. This would make it necessary to regularly monitor and evaluate the impacts of all the relevant health sector policies and programmes in order to set up essential regulatory and stabilisation mechanisms. Exploring the socioeconomic determinants requires the gradual establishment of well-specified databases and ensuring their system relations with the relevant institutions to achieve more effective policies, systems, and programmes that could be developed in such a case. The developed countries have the national registers needed to be prepared for further analytical purposes. Many countries do not have basic mortality and morbidity data stratified according to socioeconomic characteristics, which considerably limits them in the development of programmes and policies related to health equality. The international institutions that can improve local, national, and international health status and science, research, and education monitoring can play a very important supporting role.

Social stratification determines the various approaches to healthcare that result in injustice in the prevention and treatment of diseases, respectively. Health inequalities represent the result of social norms, practices, and policies that promote unfair distribution and access to wealth, power, and social resources. The intensity of their impact and the types of these policies can enlarge health inequalities. The socioeconomic determinants are the important indicators of these processes [1]. Therefore, an examination of their effects on mortality makes it possible to assess the extent of health inequalities as well as the quality and efficiency of the health system of the country [2]. These consistent facts represent also the motive for the analysis of this study, whose main aim is to examine the causal effects of the selected socioeconomic determinants of mortality in the European Union member countries [3]. Existence of significant differences between the countries is investigated so that it is possible to assess these differences in government policies among countries. This is an examination of the initial differences with the ambition of specifying the factors that are most sensitive, in order to change the mortality rate, which can also bring many interesting findings for policymakers and is important for the implementation of the outcome of comparative analyses.

## 2. Literature Review

The authors have investigated the impact of the gross domestic product and the other macroeconomic indicators on health of the population for a long time, in many studies. The choice of the macroeconomic indicators to be observed is primarily determined by the objectives of the research. The trends in these periods depending on the global economic impacts such as the economic crises or various epidemiological risks are visible. Within the countries, the tendencies to explore the impact of the macroeconomic indicators on health of the population are seen in the context of the transformation processes, the adopted reforms, the migration processes, or as a result of the various restrictive and regulatory interventions by the national governments. Despite the considerable heterogeneity of the contents of the research studies, they bring valuable findings regarding the detection of other determinants of health as well as health inequalities. They also point to the necessity of creation of national and international data collection systems and databases that possess the necessary data for the in-depth analyses required in order to construct the policymaking mechanisms in the health systems.

Reeves et al. examine the economic impact of alternative government spending through the fiscal multipliers in their study [4]. The research sample consists of the 25 European Union member countries with the period under observation from the year 1995 to the year 2010. This period also includes the global financial crisis of the years 2007 and 2008. The results of the study show that government spending can have short-term effects on population health. The authors draw attention to the fact that no research study can accurately estimate growth potential of the different types of government spending. The authors state that the effects of government spending supporting economic growth with positive fiscal multipliers, meaning return on investment, are found in the health and social protection sectors and the education sector.

Wongboonsin and Phiromswad examine a relation between the demographic structure and economic growth [5]. The authors found that the demographic structure affects economic growth differently in developed and developing countries. In developed countries, an increase in share of middle-aged workers has a positive impact on economic growth through institutions, investment, and educational channels. On the other hand, an increase in share of the senior population has a negative effect on economic growth through institutions and investment channels. For developing countries, an increase in share of young employees has a negative effect on economic growth through the investment, financial market, and trade channels.

Renton et al. analyses the World Bank Group data since the year 2005 to assess the benefits of economic growth in the particular countries [6]. The authors explore the changes of the gross domestic product by sector and whether the positive impact of the gross domestic product by sector is differentiated within the groups of the less developed and the more developed countries. The authors evaluate whether the technical progress that affects health by increasing the gross domestic product would be more beneficial for the wealthier countries than for the poorer ones, pointing out that the global health inequalities are about to increase.

The influence of health on the economic parameters of the country is also examined by Schooling et al., who implement the potential years of life lost indicator in their analyses besides the other approaches as well [7]. The authors focus on the population of the Hong Kong Special Administrative Region of the People’s Republic of China in the period beginning in the year 1976 and ending in the year 2006. The basis for their analysis lies in the premise that the structure of the diseases changes itself regarding the economic development of the countries. They examine social mortality patterns in the countries with a history of rapid economic development. In the conclusion of the study, the authors state that the social differences in health are not homogeneous, and hence, it is necessary to investigate the causes of the differences in mortality for the individual diagnoses and to implement specific prevention programmes accordingly.

Gradanos examines a relationship between the gross domestic product growth rate and the progress in health measured by increasing the life expectancy at birth [8]. The results of the analysis show that the lower the rate of for the both sexes, the stronger this effect is. It appears basically at lag zero, though some short-lag effects of the same negative sign are found.

Vandersteegen et al. introduce another determinant into their analyses that affects the health system of the country and its sustainability [9]. This factor is medical malpractice as a determinant of health spending in the Organisation for Economic Co-operation and Development countries. The authors with their analytical models prove that the health responsibility system of the country significantly influences healthcare expenditure. Therefore, they recommend to scientists and policymakers to pay more attention to detecting illicit medical practices as determinants of healthcare spending.

Meijer et al. examine the structure of healthcare expenditure and its distribution processes [10]. Attention is also paid to the examination of the management processes of healthcare facilities, in which all the processes are directly linked to financial flows, and therefore, the growth rate of the expenditures needs to be examined in detail in their individual parts.

An explanation of the importance of this topic that investigates the relationship between health spending and age together with sex through the regression analysis is also found in the study by van Baal and Wong [11]. They confirm that the growth in health expenditure is dependent on the age aspect.

The sustainability of health systems can also be explored through the indicators of lost productivity, which is associated with mortality and morbidity of the population. This is also evidenced by the study by Schofield et al., which states that it is important to investigate lost productivity—especially for the diagnoses whose treatment is for the state costliest [12]. These diagnoses include cardiovascular diseases. The authors point to several forms of the financial losses related to morbidity, which include not only lost productivity but also premature retirement due to illnesses, increased social benefits, and so forth.

According to Dahl and van der Wel, social expenditure is associated with lower health inequalities [13]. In this regard, they state that social security expenditure also needs to be examined in the context of education of the population. Based on their analysis outcome, high social transfers are mainly aimed at the inhabitants with primary education, whilst the population with higher education is not a primary addressee of these financial flows.

Braendle and Colombier assess the impact of the introduction of the new healthcare technologies on expenditure of healthcare segment [14]. In their study, they state that a higher volume of available public financial sources represents a risk of overcapacity of modern health technologies, which indicates an increase in healthcare sector spending. They consider these findings to be very important especially in decentralised and small healthcare units.

Education does not affect all the characteristics of the health system. This is also confirmed by Black et al. who demonstrate that it is not possible to make a definitive impact statement on mortality education [15].

However, the improvement of the social situation of the particular individual and employment itself are strong determinants of population health improvement as stated by James [16]. The author applied the 13 logistic regression models to predict adult mortality rates at different levels of achieved education aiming at examining ages ranging from 45 years to 84 years.

Lynch and von Hippel observe health gradient in education [17]. The authors conducted the National Longitudinal Study of Youth. The results show that more educated adult people tend to have better so-called self-rated health status. In the conclusion, the authors note that it is possible that educational attainment would have a greater effect on health at a senior age, while at a younger age, the health gradient in education is shaped primarily by selection and confounding rather than a causal effect of education on the health sector.

Buckles et al. observe the strong negative effects of higher education on the mortality by cancer and cardiovascular diseases diagnoses, which are the main causes of mortality in the elderly population [18]. The treatment of cardiovascular diseases attracts the highest volume of government expenditure as stated by Schofield et al. As long as individuals bear the economic costs of lost income, the government has to bear the direct healthcare costs. Although, this is also affected by a loss in productivity due to absence of employees from work, loss of income tax revenue, and increased government transfers aimed at aid for people.

Unemployment is also an important determinant of population mortality rate in the particular countries. Halliday investigates the relationship between unemployment and cardiovascular mortality with oncological diseases [19]. The authors state that the poor labour market conditions are associated with the higher mortality rate of the male population of productive age. The authors apply logistic regression on the sample of more than 16,000 individuals. Their outcome is that poor economic conditions bear a significant health risk.

Gravelle et al. investigate the relationship between income inequality and population health [20]. As they state in their analysis, the estimated relationship between these aspects is not significant in any of their estimated models. The authors emphasise that there are serious conceptual difficulties in usage of the aggregate cross-section data sets as the means of testing hypotheses about the effect of income with its distribution on the health status of the individuals.

Macroeconomic perspective in this field is brought by a study by Cavalieria and Ferrante [21]. The authors examine the hypothesis that a shift towards a fiscal autonomy of regional governments could improve health outcomes as measured by the infant mortality rate. The authors apply a linear fixed-effects regression model with robust standard errors covering a panel data set of the 20 regions over the period from the year 1996 to the year 2012. Higher decision-making autonomy is associated with the lower infant mortality rates, and the lower transfer dependency is related to this as well.

Karanikolos et al. examine the implications of the financial and population health crises [22]. The results of their study show that, on the one hand, increased social welfare spending significantly reduces mortality from the diseases related to social circumstances such as alcohol related death causes. On the other hand, healthcare spending does not behave in this way. Besides these factors, the number of suicides among people younger than 65 years of age has grown in the European Union since the year 2007.

Stuckler et al. examine the impact of the economic crisis on unemployment [23]. Their aim is to assess the influence of the economy changes on mortality rate and how governments could eliminate these adverse effects. The outcome of their analysis shows that each percentage point increase in unemployment rate is associated with a 0.79% rise in suicide rate for people younger than 65 years of age, although it is statistically nonsignificant. It is also accompanied by a 0.79% increase of homicide rate, whilst in contrast, road traffic causes of mortality decreased by 1.39%. A more than 3% increase in the unemployment rate has a greater impact on suicide mortality at ages younger than 65 together with mortality from alcohol abuse. The authors complete the implications of the relevant policies in a successive study [24].

Glonti et al. conduct a systematic review with the aim of examining evidence from the longitudinal studies on the factors influencing resilience for any health outcome or health behaviour among the general population living in the countries exposed to the global financial crisis of the years 2007 and 2008 [25]. The authors review the studies from the six electronic databases that are aimed at assessment of the impact of this crisis on health outcomes, health behaviour, health risks, and so forth. Ten social and demographic factors are identified in order to evaluate the level of health risks. They focus on physical health studies, mortality, suicide and suicide attempts, mental health, and health behaviour. Their results reveal that mental health of the female population is more prone to the mentioned crisis than mental health of the male population. A lower level of income is associated with a higher increase in cardiovascular diseases, higher mortality, and mental health deterioration. Employment status is related to changes in mental health. The associations with age, marital status, and education are less consistent, although higher education is associated with healthier behaviour.

Ruhm also examines the effects of the global financial crisis of the years 2007 and 2008 on mortality [26]. His analysis shows that the effects of the severe national recessions in the United States of America appear to have a beneficial effect on mortality, which is roughly twice as strong as predicted due to the increased unemployment rate itself. The higher suicide rate is related to the periods of economic recession and is predicted with a certain offset.

Conceicao et al. draw attention to the fact that there is an asymmetry in the impact of the economic crisis on health and education among the different countries seen from the perspective of their development [27]. While in the rich countries health and education results improve during the economic crisis, in the poor ones they get worse. In the conclusion, the study states that economic expansions have less significant effects than economic contractions, because recovery in human development indicators is not as rapid and steep as the deterioration that occurred during the economic downturn. There is to note also that these indicators must not be rising at all during the recovery period.

Similar results are obtained by Kim and Serra-Garcia who examine the impact of the global financial crisis of the years 2007 and 2008 on the well-being of the population in Jamaica [28]. The authors found that health and education outcomes in rich countries often improve during the economic crises, while they deteriorate in poor countries. For assessment of the macroeconomic processes, they note that economic expansions have less significant effects than economic contractions. This outcome means that recovery is not so steep as it is stated in the previously mentioned study.

Deaton concludes that the exact relationship among health, education, and economic growth is difficult to determine due to the interaction effect of the institutions and stages of human development [29,30]. In many countries, health conditions deteriorate during crises, although the results are sensitive to policy measures taken to mitigate the effects of economic crises.

## 3. Data and Methodology 

The input data come from the period beginning in the year 1992 and ending in the year 2015. The availability of the selected indicators for the examined period is negligible for all the European Union member countries.

The countries are assigned the standard geographic codes according to the International Organization for Standardization 3166 standard Codes for the representation of names of countries and their subdivisions—the two-letter alpha-2 codes: AT—the Republic of Austria, BE—the Kingdom of Belgium, BG—the Republic of Bulgaria, CY—the Republic of Cyprus, CZ—the Czech Republic, DE—the Federal Republic of Germany, DK—the Kingdom of Denmark, EE—the Republic of Estonia, ES—the Kingdom of Spain, FI—the Republic of Finland, FR—the French Republic, GB—the United Kingdom of Great Britain and Northern Ireland, GR—the Hellenic Republic, HR—the Republic of Croatia, HU—Hungary, IE—the Republic of Ireland, IT—the Italian Republic, LT—the Republic of Lithuania, LU—the Grand Duchy of Luxembourg, LV—the Republic of Latvia, MT—the Republic of Malta, NL—the Kingdom of the Netherlands, PL—the Republic of Poland, PT—the Portuguese Republic, RO—Romania, SE—the Kingdom of Sweden, SI—the Republic of Slovenia, and SK—the Slovak Republic [31].

The variables involved in the model are the following ones:-the standardised mortality rate—a number of deaths per 100,000 inhabitants according to the standard European population designed by Eurostat [32];-the gross domestic product per capita;-the total healthcare expenditures as a share of the gross domestic product;-the unemployment rate;-the healthcare system financing model.

The explained variable is represented by a standardised mortality rate. The explanatory variables perform also as the lagged variables with its current version too. This is due to the fact that the variable can develop throughout the time itself. It can influence the explained variable variously.

The regression model is in a form of a linear regression. All the employed variables are logarithmised.

A statistical significance level is set to be equal to 5% for all the regression models and all the tests.

There are the three models designed for the analysis:-the overall mortality;-the female sex mortality;-the male sex mortality.

Applying the Durbin-Wu-Hausman test, it was decided to choose the type of a panel linear regression model [33,34,35]. The overall mortality model and the male sex mortality model are suitable to employ the random effects model, whilst the female sex model is set for the fixed effects model. Because, the second model behaves in this way, it is also analysed through a random effects model approach in order to allow its comparison with the other two models.

Successively, an equality of the level constants for all the explored countries is tested and thus, the panel data structure is to be considered. For all the regression models, the p-values are lower than a 0.001 threshold. Therefore, the null hypothesis is rejected in favour of the alternative hypothesis, and that is why it is necessary to take into account the panel data structure.

The aggregate model statistics seem to be all right. For each model, as a whole, it is statistically significant. It is necessary to verify that there is no correlation between the variables involved in the regression models. The presence of this effect may cause biased estimates of regression coefficients. By conducting cross-sectional dependence tests, a problem found in the regression models can cause a misinterpretation of the outcome. Therefore, a robust matrix is employed to solve this issue. Through an application of this method, the regression coefficients are not to be changed in a misleading way. All these values are interpreted in a ceteris paribus way meaning no other variable changes throughout time in order to obtain only the influence of the sole variable. 

## 4. Analysis

The whole analysis is divided into the three main sections—the first one deals with the standardised mortality rate as the explained variable for the current period and the previous period; the second one covers the standardised mortality rate as the explained variable with the lagged variables, and thirdly, an influence of a three-per-cent increase of the unemployment rate on the standardised mortality rate regarding sex.

Figure 1 demonstrates distribution of the standardised mortality rate values among the individual countries.

### 4.1. The Total Standardised Mortality Rate Models with the Lagged Variables

The first group of the regression model defines mortality through the data of the variables from the current period together with the previous period.

#### 4.1.1. The Total Standardised Mortality Rate Model with the Lagged Variables Regardless of Sex

The very first model analyses the total mortality regardless of a sex aspect. The regression estimates are visualised in the following table.

This regression model is able to explain 79% of the variability of the explained variable. Obviously, in addition to the determinants that are considered, mortality is influenced by a number of the other factors that are not included in the regression model. One of the reasons is an absence and low availability of the data and that these do not perform as the measurable variables and factors often. All the regression coefficients except for the constant values have a negative sign. In general, the model can be considered as statistically significant. All the indicators except for a constant value negatively affect the explained variable. If the unemployment rate under the ceteris paribus condition increases by 1%, the total mortality regardless of sex decreases by 0.047%. It is seen that there is only a minimal difference between the total healthcare expenditure on health in the current period and the same expenditure in the previous period. Under the otherwise unchanged conditions, an increase in the total healthcare expenditures by 1% over a given period results in a decrease in the standardised mortality rate by 0.153%. A growth in total healthcare expenditure by 1% in the previous period under unchanged conditions has a negative impact of 0.180% in the explained variable. Under the condition of ceteris paribus, if the gross domestic product per capita increases by 1% in the current period, the standardised mortality rate decreases by 0.075%. It is visible that the gross domestic product per capita after one-year lag is the most important item in the model. An increase of this indicator by 1% causes a decrease in the standardised mortality rate by 0.110%. The last indicator is the categorical variable of the healthcare system financing model. The national health insurance system has an impact of a 0.079% drop in the explained variable compared to the social insurance system. All the selected variables have a significant impact on the overall standardised mortality rate.

#### 4.1.2. The Female Standardised Mortality Rate Model with the Lagged Variables 

The second aspect which is dealt with is the analysis of the overall mortality of the female sex. Based on the above placed Table 1 illustrating the estimation of the random effect model regression coefficients, it is seen that the model as a whole is statistically significant. It explains an eighty-per-cent share of the variability in the female standardised mortality rate. The statistically significant variables in the model are both versions of the gross domestic product per capita – the current version and also the lagged version by one period. The total healthcare expenditures with its lagged version by one period are statistically significant too. All the values of the regression coefficients are negative. Hence, these indicators negatively affect the female standardised mortality rate. Fulfilling the ceteris paribus condition, if the total healthcare expenditures related to the previous period increase by 1%, the female standardised mortality rate decreases by 0.089%. A one-per-cent increase in the female unemployment rate in the model under the unchanged conditions results in a 0.040% decrease in the explained variable. Moreover, a one-per-cent increase in the gross domestic product per capita in the current period and in the previous period causes a fall of the explained variable by 0.062% and 0.134%, respectively. It is seen that the healthcare system financing model has no significant impact on the male standardised mortality rate.

#### 4.1.3. The Male Standardised Mortality Rate Model with the Lagged Variables

The following model analyses the overall male mortality in the European Union member countries as it is illustrated by Table 1. Almost 75% of the variability of the explained variable is determined by the given regression model. All the estimates of the regression coefficients are statistically significant except for the one-year lagged gross domestic product per capita in the model. The statistical significance of the explanatory variables is confirmed—of course, with exception of the already mentioned variable. The selected indicators have a negative impact on the overall male mortality. If the male unemployment rate increases by 1% under the ceteris paribus condition, the mortality decreases by 0.056%. The total healthcare expenditure in the current period is statistically significant. Therefore, if these expenditures increase by 1%, then the male mortality decreases by 0.258%. The expenditures from the previous period are also statistically significant, and their one-percentage-point increase results in a 0.139% drop in the male mortality. The national health insurance system compared to the single-payer system has a 0.072% decrease in male mortality.

The fixed effects approach is explored also here. Based on the Durbin-Wu-Hausman test, it is appropriate to apply the fixed effects model to estimate the male standardised mortality rate. To compare the other aspects, the random effects model is applied. However, it is suitable to carry out an investigation whether the regression coefficients affecting the male standardised mortality rate develop in the same tendency as the ones from the random effects model.

Table 2 shows the estimated values of the male standardised mortality rate through the fixed effects regression model. The results are adjusted by means of a robust matrix. The model explains almost 73% of the variability of total male mortality in the European Union member countries. The model as a whole is statistically significant, and this outcome can also be applied to all the variables included in this model. The variability of examined dimensions is different from the random effects model. The gross domestic product per capita is the most important indicator in the model. The tendency of the regression coefficients moves in the same direction compared to the random effects model. There are no significant differences between the regression coefficients of both models, the random effects model and the fixed effects model.

Through the fixed effects model, the ranking of countries is created in a descending order in the following table. All the data in the table are rounded to three decimal places. Under the ceteris paribus condition, it can be expected that on average, regardless of the unemployment rate and other determinants, the male standardised mortality rate in Latvia has the highest level, peaking at 9.799. Hungary, Denmark, and Lithuania follow very closely with values of 9.752, 9.752, and 9.725. The Slovakia value lies at a level of 9.660, making it the fifth highest ranking within the European Union. Cyprus is placed in the last position with the fixed effects value lowering to 9.237. There are no significant differences between the levels of the fixed effects among the countries. The countries are sorted in a descending way in Table 3.

### 4.2. The Total Standardised Mortality Rate Models Only with the Lagged Variables

This group of the regression models examines a response of the involved variables on the explained variable.

#### 4.2.1. The Total Standardised Mortality Rate Model Only with the Lagged Variables Regardless of the Sex

This model demonstrates the impacts of the lagged variables on the standardised mortality rates for both sexes. All the variables are lagged by one period lasting one year. Almost 80% of the variability of the explained variable is clarified by the assumed regression model. The model as a whole is statistically significant. In this case, the variables with a delayed effect on mortality are considered, and all of them can be considered as statistically significant. Except for a constant value, all the regression coefficients have a negative effect on the explained variable. This means that the growth of the explanatory variable causes a decrease in the explained variable and vice versa. Under otherwise unchanged conditions, an increase in total healthcare expenditures reflecting a delayed response of 1% results in a decrease in the total standardised mortality 0.298%. Again, a one-per-cent increase in the lagged gross domestic product per capita has the effect of reducing the total mortality by 0.195%. The increase in the unemployment rate, taking into account a one-year lag, influences the mortality by 0.056%. The last variable in the model is the categorical dimension representing the healthcare system financing model. Based on the results of the random effects model, the national insurance system has a 0.066% decrease in the mortality compared to the social insurance system.

#### 4.2.2. The Female Standardised Mortality Rate Model Only with the Lagged Variables

The regression model for the female standardised mortality rate is statistically significant. Moreover, almost all the explanatory variables and the explained variable are statistically significant. There is an exception in the form of the healthcare system financing model, whose p-value slightly overcomes the five-per-cent threshold. The regression model explains almost 81% of the variability of the female standardised mortality rate. Under the otherwise unchanged conditions, a one-per-cent increase of the lagged total healthcare expenditure has an effect of reduction of the female standardised mortality rate by 0.254%. The lagged gross domestic product per capita causes a 0.204% decrease in the female mortality in the case of its one-per-cent rise. If the female unemployment rate is increased by 1%, the female mortality is reduced by 0.049%. The healthcare system financing model is a statistically insignificant variable in the model, but it has a negative influence on the explained variable.

#### 4.2.3. The Male Standardised Mortality Rate Model Only with the Lagged Variables

This section deals with the male standardised mortality rate model involving the one-year lagged variables. At first sight, the model itself is statistically significant. All the variables included in the model are statistically significant. The chosen model explains almost 76% of the variability of the male standardised mortality rate. It can be seen that all the selected indicators negatively influence the explained variable. Under the ceteris paribus condition, the total healthcare expenditures from the previous period increased by 1%, causing a decrease in the male standardised mortality rate to a level of 0.352%. A growth of the lagged gross domestic product per capita by 1% has the effect of reducing mortality by 0.192%. An increase in the male unemployment rate by 1% results in a decrease in mortality by 0.056%. The national health system financed by government compared to the single-payer system decreases male mortality by 0.091%.

As in the previous case, the fixed effects approach is employed here too. Applying the Durbin-Wu-Hausman test, it is found that in case of testing the male standardised mortality rate model regarding a one-year lag, it is appropriate to use the fixed effects model. In order to compare the results with the other regression models, the random effects model is also executed. In this case, the fixed effects model is introduced, and it is compared whether the signs of the regression coefficients are identical to the random effects model.

Table 4 displays the estimated coefficients through the fixed effects model.

The fixed effect model determines 73% of the male mortality rate. The difference between the variability explained by the fixed effects model and the random effects model is minimal. It is seen that all the indicators except for the healthcare system financing model are statistically significant. In contrast to the random effects model, just right, this indicator makes a distinction. The signs of all the variables are the same as in the previous case, meaning a negative impact. The difference between the values of the regression coefficients estimated through the random effects model and the fixed effects model is negligible. Table 5 illustrates the country ranking of the fixed effects model ordered in a descending way.

In a one-year lagged model, it can be expected that on average, under the otherwise unchanged conditions, irrespective of a level of the other indicators, the male standardised mortality rate in Latvia be the highest one peaking at 9.843. The Slovak Republic is ranked sixth with its value of 9.712. The country with the lowest mortality is Cyprus with a value lowering to 9.277. Based on this outcome, there are no significant differences between the values of the fixed effects of the individual countries in the regression model.

### 4.3. The Unemployment Rate Increase Standardised Mortality Rate Models

An impact of a sharp increase in unemployment rate is found on the overall mortality in the European Union member countries together with the selected determinants in terms of the various causes, including sex. An idea about this lies in the study by Stuckler and McKee, who analyse the effects of the crisis on the overall health status of the European Union population and also the responses of the governments. They criticise many studies that examine this fact by means of the gross domestic product indicator or the specific periods considered to be a crisis. In their study, they state that a different period may be critical for each country. They deliberate over the fact that the unemployment rate in the explored area rose by more than 3% in the critical period. We have implemented this interval into the model along with the already mentioned selected indicators. There are many other suitable indicators, but because of usage of a cross-sectional view of all the European Union member countries, there is a great lack of the suitable data, either in the terms of the explored period or low country transparency and so forth. Here, the logarithmic regression is applied too. Moreover, a one-year response approach is taken into account.

If taking a look at the incidence of mortality due to ischemic heart diseases, it is seen that mortality due to this cause is more prevalent for the male sex. In the long term, the standardised mortality rate has a decreasing tendency for both sexes. In the year 1993, the female sex had a level of 100.8 and the male sex a level of 206.2, which represents the highest number of the male standardised mortality rate in the explored period. Mortality for both sexes decreased by more than half throughout the observed period.

As for the standardised mortality rate for suicide, it is again more significant in the case of the male sex than the female sex. The numbers of suicides committed throughout the explored period kept a decreasing tendency. The difference is particularly noticeable when looking at the beginning of the period. The highest figure for the male sex is found at a level of 21.63 in the year 1993. On the other hand, the highest rate for the female sex peaked at 6.57 in the year 1992.

Table 6 demonstrates the regression coefficients all the random effects regression models according to the groups of the diagnoses.

#### 4.3.1. The Unemployment Rate Increase Total Standardised Mortality Rate Model Regardless of Sex

The general standardised mortality rate regression model seems to be statistically significant. It determines almost 79% of the variability of the explained variable. All the involved variables with exception of unemployment rate growth, which fulfils a ten-per-cent threshold, are statistically significant too. Under the ceteris paribus condition, an increase in the total healthcare expenditures from the previous period by 1% results in a decrease in the standardised mortality rate of 0.339%. The one-period lagged gross domestic product per capita growth has the effect of decreasing the total mortality by 0.182%. The national health system has a 0.081% decrease in mortality compared to the single-payer system. A three-per-cent increase in the unemployment rate has only little statistically significant impact, but it has a positive sign. Hence, its presence supports an increase of the standardised mortality rate.

#### 4.3.2. The Unemployment Rate Increase Total Standardised Mortality Rate Model Regardless of Sex on the Ischaemic Heart Diseases

The second aspect of mortality examined regardless of the sex of the population cause of mortality due to the ischemic heart diseases. The outcome values of the regression coefficients have a trend in a similar way as the general regression model mentioned previously. The ischaemic heart diseases regression model determines 62% of the variability of the explained variable. The whole model with all its explanatory variables is statistically significant. The healthcare system financing model performs as an insignificant variable. All the remaining variables are significant. They have a negative effect on mortality. Under the otherwise unchanged conditions, an increase in the total healthcare expenditures from the previous period by 1% has the effect of reducing the mortality rate for this cause by 0.824%. The gross domestic product per capita growth of 1% results in a 0.277% decrease in mortality.

#### 4.3.3. The Unemployment Rate Increase Total Standardised Mortality Rate Model Regardless of Sex in Suicide Diagnoses

Another group of causes of mortality which is dealt with is a set of the suicide diagnoses. The model explains little less at a level of 30% of the standardised mortality rate. Certainly, there is plenty of other factors that would need to be included in the regression model, and especially in this case, from a psychological point of view, it should be considered too. However, there is a low availability of data for such explored period. Secondly, these are often immeasurable dimensions that can influence mortality. The model itself and all the variables except for the total healthcare expenditures are statistically significant. This explanatory variable has a positive sign. Mortality rises by 0.233%, if the gross domestic product per capita growth is increased by 1%. The healthcare system financing model has the effect of reducing the explained variable by 0.206%. In this section, the most important dimension is in the unemployment rate variable that is statistically significant. Under the otherwise unchanged conditions, an increase in the unemployment rate by 3% has a positive impact on the mortality by 0.087%.

#### 4.3.4. The Unemployment Rate Increase Female Standardised Mortality Rate Model

A similar succession of the analytical steps is carried out also for both sexes. Firstly, the female sex regression model is analysed as seen in Table 7.

The main outcome of this section is that the regression model along with all its variables is statistically significant except for the unemployment rate growth having a positive sign. The model explains almost 80% of the variability of the explained variable. A total healthcare expenditure increase of 1% causes a reduction in the overall female mortality of 0.278% under the ceteris paribus condition. Gross domestic product per capita growth of 1% results in a 0.192% decrease in mortality and the national health system compared to the single-payer system shows a 0.055% decrease.

#### 4.3.5. The Unemployment Rate Increase Female Standardised Mortality Rate Model on the Ischaemic Heart Diseases

The ischaemic heart diseases model as a whole is statistically significant. However, the healthcare system financing model variable is statistically insignificant, and the unemployment rate growth fulfils only a ten-per-cent significance threshold. Whilst the first mentioned variable influences the explained variable in a positive way, the latter one does so in a negative way. A share of 60% of female mortality variability is explained by the regression model. An increase in the total healthcare expenditure of 1% has the effect of a 0.843% decrease in female mortality together with a decrease in gross domestic product per capita growth of 0.293%. 

#### 4.3.6. The Unemployment Rate Increase Female Standardised Mortality Rate Model on the Suicide Diagnoses

The regression model is statistically significant itself. The variables except for the total healthcare expenditures and the unemployment rate growth are statistically insignificant and both bear a negative impact on female mortality. The defined regression model explains 26% of the mortality variability. A one-per-cent increase in the gross domestic product per capita growth results in a decrease in suicide mortality of 0.288%. The national insurance system compared to the social insurance system has an effect on the reduction of mortality of 0.279%.

#### 4.3.7. The Unemployment Rate Increase Male Standardised Mortality Rate Model

The same succession of analytical steps is carried out for the male sex. Table 8 introduces the figures for the male sex regression models.

All the variables with the regression model itself are statistically significant except for the unemployment rate growth variable. The model explains almost 75% of the male mortality. The unemployment rate growth also keeps a positive influence on mortality here. If the total healthcare expenditure under the ceteris paribus condition rises by 1%, a decrease of 0.405% in the examined mortality occurs. If such an increase in the gross domestic product per capita growth happens, it reduces the mortality by 0.179%. The healthcare system financing model has an effect by way of a decrease of 0.105%.

#### 4.3.8. The Unemployment Rate Increase Male Standardised Mortality Rate Model on the Ischaemic Heart Diseases

The regression model explains almost 62% of the variability of male mortality. The model as a whole is statistically significant with the involved variables except for the healthcare system financing model and the unemployment rate growth. The first one influences the explained variable in a negative way and the second one in a positive way. Under the ceteris paribus condition, an increase in the total healthcare expenditures of 1% is reflected in a 0.839% decrease of the examined mortality. The gross domestic product per capita growth of 1% effects a 0.065% reduction in male mortality.

#### 4.3.9. The Unemployment Rate Increase Male Standardised Mortality Rate Model on the Suicide Diagnoses

A last examined view is the suicide aspect for the male sex. The assigned regression model clarifies almost 28% of male mortality variability. The origin of this low figure lies in the fact that suicides are also influenced by other various non-measurable reasons. This regression model as a whole is statistically significant with the involved variables except for total healthcare expenditures having a positive influence. An increase in the gross domestic product per capita growth of 1% causes a decrease in mortality of 0.227%. The healthcare system financing model effects a 0.204% decrease of the explored mortality. The unemployment rate growth has a positive impact at a level of 0.117%.

## 5. Discussion

One of the important indicators of population health is mortality, which has explicit relation not only with the social and health systems but also with economic parameters. Although the health of the population has been improved in the different countries of the world in a long-term perspective, the health inequalities persist, not only between the countries but also within the countries [36,37]. The health inequalities are not only unfair but also economically and socially extremely costly [38]. For this reason, it is very important to examine the socioeconomic determinants of health in the working and living conditions of the population of the explored countries in order to quantify their impact on life expectancy as well as on productivity levels and healthcare spending. Many studies report the negative impacts of the adverse economic and social situation of countries on the health of the population. Investigation of its consequences—for instance, long-term unemployment and poverty for the health of the population and a possibility of creating affordable social protection with effective safety networks to eliminate these negative impacts. The chronic noncommunicable diseases and their development in the individual countries are explored very weakly globally despite strong evidence of the negative health impacts on individuals as well as on society [23,39,40]. Many international research teams examine population trends in chronic noncommunicable diseases using theoretical and empirical models to quantify some of the economic effects of their growth rate. They apply the experience coming from developed countries throughout the analytical processes. The availability of the relevant data also allows a deeper examination of the factors causing health progress, whose main causal variables could be economic development, public health financing, health infrastructure, and access to health care [24,41]. This is reflected in the creation of multidimensional research studies, the implications of which are translated not only into health and social policies but also into economic ones. Another benefit of these studies is the gradual creation of international benchmarks and the implementation of research findings into active development programmes. The aforementioned facts are also followed by this research study, which aims to investigate the socioeconomic determinants of health of the mortality of the population of the European Union member countries.

The whole paper is divided into the several segments as the analysis flow continues. There are several findings from the perspective of the individual countries stated in the text, but they should be considered through the observed dimensions in the regression models. In order to obtain as much information as possible, the separate regression models of the different types are applied. Firstly, the random effects model modelling the standardised mortality rate creates a basis for the whole analysis. It is visible that all the dimensions have a negative influence on the explained variable. The unemployment rate under the ceteris paribus condition increased by 1% brings the total mortality, regardless of sex, down by 0.047%. The following variables involved in the model cause successive impacts—the total healthcare expenditure decreases at a level of 0.153% with its lagged version at a level of 0.180%; the gross domestic product decreases at a level of 0.075% with its lagged version at a level of 0.110%, and finally, the national health insurance system decreases at a level of 0.079%. A more interesting fact is that the lagged variables of the gross domestic product and the total healthcare expenditures have a higher negative influence of 46.6% and 17.65%. It is clearly demonstrated that the longer period causes a more negative impact. On the other hand, the fixed effects model concentrating on the impact for the particular countries reveals that the lagged variables have lower negative influence. In the case of the gross domestic product, it is 5.43%, whilst the total healthcare expenditure has a higher decrease in its influence at a level of 12.15%. From the territorial point of view, Latvia has the highest level of fixed effects, meaning the highest constant value for the base explained variable, which is 5.74% higher than the lowest value assigned to Cyprus. In the case of the regression model with the only lagged variables, the countries on the extreme points are the same, but the fix effects values are higher in an absolute way, although in a relative way, their difference is almost the same at a level of 5.75%. From an overall view, the only lagged variables regression models behave with stronger influence on the explained variable representing the standardised mortality rate.

The essential regression model without distinction of sex behaves as follows—the decreases caused by the explaining variables are at a level of 0.075 in the case of the gross domestic product, with 0.110 in the case of its lagged version; 0.153 in the case of the total healthcare expenditure, with 0.180 in the case of its lagged version; 0.047 in the case of the unemployment rate and, finally, 0.079 in the case of the healthcare system financing model. The female alternative of the previous regression model states that the female standardised mortality rate decreased by 0.188 in the case of lagged total healthcare expenditure, by 0.040% in the case of the female unemployment rate, and by 0.062% in the case of the gross domestic product per capita growth with 0.134% in a case of its lagged version. On the other hand, the male standardised mortality rate regression rate records decreases for all the observed variables—the unemployment rate to a level of 0.056%, the total healthcare expenditures to a level of 0.258% with its lagged version to a level of 0.139% and, finally, the healthcare system financing to a level of 0.072%. The lagged version of the female standardised mortality rate regression model expresses the following reductions of the explored mortality by a one-per-cent increase of the explaining variables—through the total healthcare expenditures by 0.254%, the gross domestic product per capita by 0.204%, and the female unemployment rate by 0.049%. The male version of this model enhances one more variable—the explored mortality decreased through the total healthcare expenditures by 0.352%, the gross domestic product per capita by 0.192%, the male unemployment rate by 0.056%, and the healthcare system financing by 0.091%. The unemployment rate increase regression model for the standardised mortality rate of both sexes brings decrease for all the variables—the total healthcare expenditures by 0.339%, the lagged gross domestic product per capita growth by 0.182%, the healthcare system financing by 0.081%, whilst the essential three-per-cent increase of the unemployment rate has a positive effect. The negative impacts are repeated also in a case of the ischaemic heart diseases regression model for both sexes, where the total healthcare expenditure causes a decrease of 0.824% and the gross domestic product per capita growth of 0.277%. Moreover, the suicide diagnoses regression model behaves in the same way—the total healthcare expenditure decreases the observed mortality rate by 0.824%, the gross domestic product per capita growth by 0.277% decrease in the mortality. The female alternatives of the previous three models have a very similar behaviour. The overall unemployment rate increase model brings a decrease for all the examined explaining variables—the total healthcare expenditure by 0.278%, the gross domestic product per capita growth by 0.192%, and the healthcare system financing by 0.055%. In the case of ischaemic heart diseases, the reductions are higher, as the total healthcare expenditure decreases the explored mortality rate by 0.843% and the gross domestic product per capita growth by 0.293%. The suicide diagnoses regression model has a 0.288% decrease caused by the gross domestic product per capita growth and a 0.279% decrease caused by the healthcare system financing. The male alternatives of the unemployment rate regression models bring the following results. The male standardised mortality rate is decreased by the total healthcare expenditures to a level of 0.405%, the gross domestic product per capita growth to a level of 0.179%, and the healthcare system financing model to a level of 0.105%. The ischaemic heart diseases regression model records decreases by total healthcare expenditure of 0.839% and gross domestic product per capita growth of 0.065%. Finally, the suicide diagnoses regression model encompasses the following impacts: a 0.227% decrease by the gross domestic product per capita growth, a 0.204% decrease by the healthcare system financing model, and a 0.117% increase by the unemployment rate growth.

As it is seen, all the regression models behave in a considerable pattern that is recognisable through all the impacts of the particular explained variables. Almost all the examined cases are represented by the negative impact to the explained standardised mortality rate—it does not matter whether it is the overall standardised mortality rate or the ischaemic heart diseases standardised mortality rate, or the suicide diagnoses standardised mortality rate. To summarise all the findings, it has to be noted that the analysis outcome is not comprehensive, and hence, it creates a potential platform for the following research.

## 6. Conclusions

Economic growth is important not only for developed countries but also for developing ones, as it gives them the opportunity to raise resources and to invest in improving the lives of their population. An important indicator representing the economic growth of countries is the gross domestic product. Economic performance measured as the gross domestic product per capita varies across the European Union member countries with significant income inequalities persisting in the long run. An examination of the links between this indicator and the level of health of the population expressed through population mortality provides an initial picture of the impact of the economic parameters on the health status of the population. In addition to this context, it is necessary to monitor the impact of the policies and the socioeconomic indicators on the health of the population and thus to identify the causes of the emerging disparities not only within the country but also between the countries and among the countries. The health disparities arise partially as a result of the circumstances and conditions in which people grow up, live, work, and age. They are also affected by the availability of health care provision, the volume of health system expenditures, and the health behaviour of the population. The aim of the study is to examine the impact of the socioeconomic determinants on the health of the population of the European Union member countries. The outcome of the analysis brings up interesting findings about the individual dimensions affecting the standardised mortality rate for both sexes. The results of the study support the creation of national policies that should be oriented towards the elimination of health inequalities, not only within the individual countries but also from an international point of view. In the field of public health, several research projects could be initiated, which would intensify the evaluation of the causal relationships and impacts of the influences of the socioeconomic determinants on the social and economic spheres of the countries.

## Figures and Tables

**Figure 1 ijerph-17-04699-f001:**
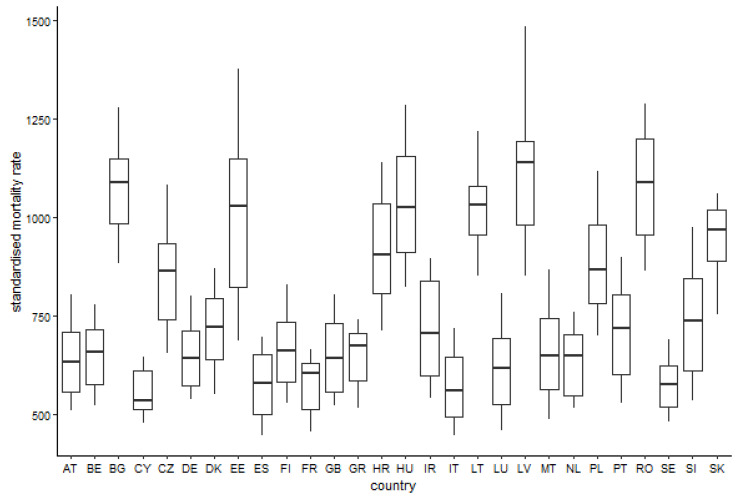
Boxplot of the standardised mortality rate for the individual countries.

**Table 1 ijerph-17-04699-t001:** The random effects models with the lagged variables.

Variable	Overall Mortality		Female Sex Mortality		Male Sex Mortality	
	Regression Coefficient	*p*-Value	Regression Coefficient	*p*-Value	Regression Coefficient	*p*-Value
C	9.204	0.000	8.914	0.000	9.586	0.000
GDP_t_	−0.075	0.005	−0.062	0.017	−0.148	0.000
GDP_t–1_	−0.110	0.000	−0.134	0.000	−0.033	0.142
E_t_	−0.153	0.004	−0.089	0.064	−0.258	0.000
E_t–1_	−0.180	0.005	−0.188	0.003	−0.139	0.037
UR	−0.047	0.004	−0.040	0.028	−0.056	0.002
FM	−0.079	0.002	−0.052	0.058	−0.072	0.001

Source: own elaboration by the authors.

**Table 2 ijerph-17-04699-t002:** The fixed effects model with the lagged variables.

Variable	Regression Coefficient	*p*-Value
log(GDP_t_)	−0.092	0.003
log(GDP_t–1_)	−0.087	0.000
log(E_t_)	−0.214	0.002
log(E_t–1_)	−0.188	0.013
log(UR)	−0.051	0.003
log(FM)	−0.073	0.010

Source: own elaboration by the authors.

**Table 3 ijerph-17-04699-t003:** The fixed effects of the individual countries in the model with the lagged variables.

Country	Fixed Effects Coefficient
Latvia	9.799
Hungary	9.752
Denmark	9.725
Lithuania	9.720
Slovakia	9.660
Finland	9.658
Estonia	9.653
Germany	9.606
Croatia	9.604
Ireland	9.595
Austria	9.580
Portugal	9.564
Belgium	9.558
Slovenia	9.540
France	9.529
United Kingdom of Great Britain and Northern Ireland	9.517
Poland	9.515
Sweden	9.505
Bulgaria	9.502
Czechia	9.492
Luxembourg	9.487
Netherlands	9.485
Spain	9.443
Greece	9.441
Italy	9.436
Romania	9.360
Malta	9.356
Cyprus	9.237

Source: own elaboration by the authors.

**Table 4 ijerph-17-04699-t004:** The fixed effects model only with the lagged variables.

Variable	Regression Coefficient	*p*-Value
log(GDP_t–1_)	−0.194	0.000
log(E_t–1_)	−0.348	0.000
log(UR_t–1_)	−0.060	0.001
log(FM_t–1_)	−0.048	0.056

Source: own elaboration by the authors.

**Table 5 ijerph-17-04699-t005:** Fixed effects of the individual countries in the model only with the lagged variables.

Country	Fixed Effects Coefficient
Latvia	9.843
Hungary	9.791
Lithuania	9.771
Denmark	9.747
Estonia	9.713
Slovakia	9.712
Finland	9.698
Germany	9.651
Croatia	9.650
Ireland	9.637
Austria	9.619
Belgium	9.607
Slovenia	9.577
France	9.576
Portugal	9.576
Poland	9.565
United Kingdom of Great Britain and Northern Ireland	9.551
Luxembourg	9.546
Bulgaria	9.541
Sweden	9.536
Netherlands	9.531
Czechia	9.530
Spain	9.473
Italy	9.464
Greece	9.458
Romania	9.410
Malta	9.376
Cyprus	9.277

Source: own elaboration by the authors.

**Table 6 ijerph-17-04699-t006:** The random effects models according to the groups of the diagnoses.

Variable	Overall Mortality		Suicide Diagnoses		Ischaemic Heart Diseases	
	Regression Coefficient	*p*-Value	Regression Coefficient	*p*-Value	Regression Coefficient	*p*-Value
C	9.080	0.000	4.607	0.000	9.132	0.000
GDP_t–1_	−0.182	0.000	−0.233	0.000	−0.277	0.000
E_t__–__1_	−0.339	0.000	0.099	0.176	−0.824	0.000
UR_t–1_	0.026	0.093	0.087	0.019	0.063	0.095
FM_t–1_	−0.081	0.003	−0.206	0.009	−0.052	0.515

Source: own elaboration by the authors.

**Table 7 ijerph-17-04699-t007:** The random effects models according to the groups of diagnoses for the female sex.

Variable	Overall Mortality		Suicide Diagnoses		Ischaemic Heart Diseases	
	Regression Coefficient	*p*-Value	Regression Coefficient	*p*-Value	Regression Coefficient	*p*-Value
C	8.787	0.000	4.540	0.000	8.949	0.000
GDP_t–1_	−0.192	0.000	−0.288	0.000	−0.293	0.000
E_t–1_	−0.278	0.000	−0.023	0.831	−0.843	0.000
UR_t–1_	0.022	0.157	−0.033	0.563	0.074	0.093
FM_t–1_	−0.055	0.035	−0.279	0.009	−0.045	0.604

Source: own elaboration by the authors.

**Table 8 ijerph-17-04699-t008:** The random effects models according to the groups of the diagnoses for the male sex.

Variable	Overall Mortality		Suicide Diagnoses		Ischaemic Heart Diseases	
	Regression Coefficient	*p*-Value	Regression Coefficient	*p*-Value	Regression Coefficient	*p*-Value
C	9.452	0.000	5.006	0.000	9.424	0.000
GDP_t–1_	−0.179	0.000	−0.227	0.000	−0.269	0.000
E_t–1_	−0.405	0.000	0.115	0.124	−0.839	0.000
UR_t–1_	0.025	0.127	0.117	0.001	0.052	0.140
FM_t–1_	−0.105	0.001	−0.204	0.012	−0.065	0.407

Source: own elaboration by the authors.

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
