# Peer review of "Exploration of Influence of Socioeconomic Determinants on Mortality in the European Union"

_ijerph, 2020, doi:10.3390/ijerph17134699_

Round 1

Reviewer 1 Report

Page 2 line 58: “creating the national productivity” does not make sense grammatically. There are numerous instances where English grammar should be address such that verb tenses agree within sentences.

Page 2 line 67-68 “The health disparities should be one of the main indicators of state performance” This statement requires a great deal more justification and specificity.

The authors do not provide a clear justification as to why lagged models are important or change reader's understanding of results.

Result tables are underdeveloped. Typically, these include t/z-values at the minimum and regularly include confidence intervals for estimates.

Define Ceteris paribus for your reader.

The authors are claiming that as unemployment increases, mortality decreases. This requires substantial explanation. Why does it flip in other models?

Have the authors corrected p-values for multiple testing? Why or why not?

The discussion includes much of the same information as the results. The manuscript could be much shorter and include all the relevant information for readers. The authors go through a long process of restating the findings in the discussion section but do not offer enough in the way of interpreting what is different across models and why your reader should care. They also don't tell us why it is necessary to estimate so many models.

For instance, they state that it is interesting where lagged and contemporaneous models disagree but provide not interpretation of what that means or why it is interesting. What does this tell us about the relationship between socioeconomic indicators and population health.

Author Response

Dear reviewer
Thank you very much for your valuable comments. We hope that these revisions improve the paper so that you now deem it worthy of publication in International Journal of Environmental Research and Public Health and our revision has improved the paper to a level of your satisfaction. We have made the following changes in the text according to your suggestions:
1. It is a mistake – it should be influencing instead creating.
2. We have edited in order not so strongly to point to the state of the population health.
3. More explanation was added to the application of the lagged regression models.
4. We have excluded the t-test statistic values because the p-values come from these values directly. And another point is that there are many regression models and the whole paper contains 8 tables with the models, so it would be at least doubled in order to involve also t-values. Therefore, we offer such view on the regression models. Finally, the p-value demonstrate the most important point, that is, statistical significance of the explanatory variable.
5. Definition of ceteris paribus is added. But it is so general term that it is common to use it without explanation, though in a field of economics.
6. This is good point but it is for other study rather. It could have many explanations and for one paper it is more that it could be involved in the analysis. So far, this paper is 21 pages long and we would like to keep the attention of reader to the core of the analysis.
7. Probably, we do not understand properly what do you mean by this. What do you consider as corrected p-value?
8. Yes, you are right. Originally, not so many numbers were involved in the discussion, but previously we were requsted to add such information to the dicussion. If it was neccessary, we would leave these pieces of information in order to shorten the manuscript.
9. This point is related partially to the sixth point. It is more complex problem. But such a comprehensive analysis is difficult significantly. Not only to carry it out, but also to interpret all the obtained values.

Reviewer 2 Report

Review Report

This reviewer commends the authors for their study titled, “Exploration of Influence of Socioeconomic  Determinants on Mortality in the European Union.”

The following questions and comments need addressing:

  1. It is interesting to note that the authors have provided zero in-text citations and zero references for their entire introduction. The Introduction and Literature Review should be abridged and merged into one Section:  Introduction and appropriate in-text citations and references need to be provided as may be appropriate.  Introduction and Review of Literature presented separately may be appropriate for a Thesis/Dissertation paper, as it appears to be the case here.
  2. What is/are the Objectives of the study? the objective (s) of the study should be clearly (re)articulated.
  3. What is the knowledge gap this study tries to bridge?
  4. What is the new information/evidence/knowledge that his study is attempting to produce?
  5. What is the research design?
  6. What is/are the research questions(s)?
  7. What is /are the independent (outcome) variable(s)?
  8. What are the dependent variables?
  9. What is the sampling unit, sampling method, sampling type, and sample size?
  10. What is the data analysis plan/method? What software were used in data analysis?
  11. What are the social determinants found to be statistically significant for each of the following outcomes: the overall mortality rate, and gender-specific mortality rates
  12. The authors report that the input data come from the period beginning in the year 1992 and ending in the year 2015. Social determinants (education, income, unemployment, etc. );  government policy; macroeconomics, and social structure have changed  significantly over the  last  three to four decades.  How do the authors justify evidence produced from collected data as old as four decades ago can be made to reflect current European public health reality and be used to make policy decisions  or public health interventions aimed at, for instance, achieving sustainable development goals in the respective countries across European Union?
  13. There is very little  narrative devoted to the comparison of the present study with other studies and a paucity of references for such comparison in the discussion section.
  14. What are the limitations of this study?

Author Response

Dear reviewer

Thank you very much for your valuable comments. We hope that these revisions improve the paper so that you now deem it worthy of publication in International Journal of Environmental Research and Public Health and our revision has improved the paper to a level of your satisfaction. We have made the following changes in the text according to your suggestions:

1. We could join together these two sections. But we selected this way, because firstly, we would like to introduce the discussed topic to reader generally and subsequently, secondly, we would like to get reader into the theoretical background of the topic. Therefore, we have distinguished these two parts. If it was neccessary, we could join them. Another point is to make all the sections clear for reader and hence, we divided it into the two parts.
2. Social stratification determines the various approaches to healthcare that results in injustice in the prevention and treatment of the diseases, respectively. The health inequalities represent the result of the social norms, practices and policies that promote unfair distribution and access to wealth, power and social resources. The intensity of their impact and the types of these policies can enlarge the health inequalities. The socioeconomic determinants are the important indicators of these processes. Therefore, an examining of their effects on mortality make it possible to assess the extent of the health inequalities as well as the quality and efficiency of the health system of the country. These consistent facts represent also the motive for the analysis of this study, whose main aim is to examine the causal effects of the selected socioeconomic determinants of mortality in the European countries. An existence of the significant differences between the countries is investigated, so that it is possible to assess these differences in the government policies among the countries. It is an examination of the initial differences with an ambition to specify the factors that are most sensitive to change the mortality rate values which can bring many interesting findings for policymakers and it is important for the implementation of the outcome of the comparative analyses too. These ones will also be important in setting up processes to eliminate the health inequalities between the countries declared by the major international institutions.
3. It is quite difficult to answer this question. The socio-economic inequalities in health are still a topical issue and persist in the developed countries too. They represent one of the most serious and most unfair forms of social inequality and they bring great social costs at the same time. There is a close link between the health inequalities and the overall health status of the population – the greater the health inequality, the worse the health of the country's population as a whole. The presence of the socio-economic inequalities in health is one of the well-documented and repeatedly confirmed research hypotheses, despite the high variability in measurement approaches. Sociological research, which plays one of the most important roles in this area, rarely analyses the objective indicators of health. A more frequently used way is a subjective indication, which relates to the presence of the disease, health restrictions and the assessment of one's own health or the determination of the degree of satisfaction. Therefore, subjective health assessment belongs to the standardly used proxy indicators of objective health with confirmed strong validity and reliability. Many studies examining the influence of socio-economic determinants on subjective health assessments work with only simple indicator, whilst working with aggregated indicators represents still an exception. This is also due to methodological complexity, while the quality and completeness of international databases is also important.
4. After completing the paper, we have found out that this is issue so complex, it would employ a very comprehensive study to cover all the examined field.
5. The research design is based on the common kind of the analysis of sensitivitiy in a form of the regression analysis. We decided that an application of the standard models would bring the interesting results altogether with the lagged regression models. They are specific in revealing of the delayed response in a relation to the explained variable.
6. The main aim of the paper is to create a basement for further research, because it is a very comprehensive topic, it is impossible to cover all the aspects of the examined dimensions. Therefore, the main research questions are rather general views on the explored variables. We have added the paragraph involving aiming of the analysis at the end of the Introduction section.
7. The independent variables are the values of the standardised mortality rate. We distinguish the overall mortality for the both sexes, the standardised mortality rate for the female sex, and the standardised mortality rate for the male sex. It is very important, because the both sexes have the different predispositions for the particular diagnoses. It is common approach absolutely.
8. The dependent variables, that is, the explanatory variables are the gross domestic product per capita, the total healthcare expenditures as a share of the gross domestic product, an unemployment rate, and the healthcare system financing model. They are listed in the Data and Methodology section.
9. Sampling unit is defined according to the explained variable, that is, the standardised mortality rate – a number of deaths per 100,000 inhabitants according to the standard European population designed by Eurostat. It is very important to keep this in order to be compared with the other studies. The employed sampling method is the standard methodology in a form of the sensitivity analysis. The sample size is defined according to the accessible data through the common public database. We have covered the time period beginning in 1992 and ending in 2015.
10. The R software environment was employed in order to carry out the analysis. It is very good statistical software to perform whatever analysis.
11. This question is very complex and to answer it, it requires a very comprehensive reply. It depends on an angle of view, because each aspect can bring up little different answer with alternation of the independent social determinants.
12. We have not applied the four decades old data because it would be quite uninterpretable. But the longer examined period is, the more acceptable it is. The twenty-three-year time span can serve as a good basement for the exploration of the influences of the particular explanatory variables.
13. Yes, this could be point, but we have tried to find the similar studies and we would be very glad if our study can serve as an example for further research in other territories or for similar socioeconomic research.
14. The limitations are present in a form of the access to the data mainly. This is the most considered issue in this point. If the data were complete with the longest history as possible, it would allow the deeper analytical processes in order to obtain the more exact outcome.

Reviewer 3 Report

My major recommendations for the improvement of the manuscript are:

In my opiniot the observation that Stuckler examines the impact of the economic crisis on unemployment should be used to a larger and broader discussion about that inluence on the main paper subject i.e. socioeconomic determinants influence on mortality.

I also recommend that Authors can use some more literatuse from Central and V4 literature about mortality and it influence on public health quality.

Author Response

Dear reviewer
Thank you very much for your valuable comments. We hope that these revisions improve the paper so that you now deem it worthy of publication in International Journal of Environmental Research and Public Health and our revision has improved the paper to a level of your satisfaction. We have made the following changes in the text according to your suggestions:
1. Yes, you are right this could cover more fields as they are examined in the analysis. But this topic is very comprehensive. It could be very difficult to explore more dimensions. Even, now in this analysis, it is questionable what dimension has more influential power on the explanatory variable. But, we have tried to interpret as many points as possible.
2. It is good point, but it is difficult to find such studies from this area. Hence, we have point our attention also to the other territories. In the further research, we will try to do this more deeply in the V4 area.

Reviewer 4 Report

Dear authors,

thank you very much for the opportunity to review your submission.

The paper is well written, only minor changes are suggested:

The introduction lacks of any citation for numerous statements made that may be non evident to all readers of the paper, please add citations where appropriate to justify these statements

Some minor typos are still in this paper (e.g. line 228 Ruhm examines ).

Reference list does not comply to instructions for authors (e.g. no ISSN, no online references), please ammend.

Good luck!

Author Response

Dear reviewer
Thank you very much for your valuable comments. We hope that these revisions improve the paper so that you now deem it worthy of publication in International Journal of Environmental Research and Public Health and our revision has improved the paper to a level of your satisfaction. We have made the following changes in the text according to your suggestions:
1. The references were added to the Introduction section.
2. This typo and also a few others were corrected.
3. The ISSN numbers were added to all the references.

Round 2

Reviewer 1 Report

4. We have excluded the t-test statistic values because the p-values come from these values directly. And another point is that there are many regression models and the whole paper contains 8 tables with the models, so it would be at least doubled in order to involve also t-values. Therefore, we offer such view on the regression models. Finally, the p-value demonstrate the most important point, that is, statistical significance of the explanatory variable.

Yes, this is the point, there are too many models, the narrative doesn't bind them together and the authors do not provide enough information for a knowledgeable reader to assess their method. This may be a disciplinary difference but in my discipline, reporting only p-values is highly frowned upon.

7. Probably, we do not understand properly what do you mean by this. What do you consider as corrected p-value?

Conducting many statistical tests will result in Type I error. Alpha levels are corrected as such to avoid Type I error (Bonferroni correction for example). This is common practice in my discipline to improve rigor and avoid capitalizing on chance.

9. This point is related partially to the sixth point. It is more complex problem. But such a comprehensive analysis is difficult significantly. Not only to carry it out, but also to interpret all the obtained values.

It remains unclear what the reader should take away from the analysis presented here. That is the most important aspect of the paper.

Author Response

Da reviewer

We appreciate your valuable comments. Thank you very much for your reply.

Here, we offer the following suggestions:
4. Perhaps, it is also a point of the discipline or a field itself. But economics and mathematics are exact disciplines and because of the fact that p-values are firmly related to t-values, there is no doubt about their interpretation.
7. As type I error means the rejection of a null hypothesis that is true, we have treated it through an inspection of the p-values. It is possible to carry out the further tests, but we think this would put more burden on reader than it brings benefit. We try to extract all the redundant statistical tests and all the figures that could be misleading for common reader. This is not paper aimed at the mathematical audience. We try to elucidate the analysis outcome for as broad audience as it is possible.
Bonferroni correction is a well-designed approach, but it is rather for more complicated models. We have standard analysis of sensitivity here and its regression models. For a usual level, it is enough to use such application.
9. The main outcome is a possibility to create a platform for further research. This paper has not an ambition to be the desired individual analytical paper. It tries rather to demonstrate it is available to prepare a platform created of not only the data itself, but also of the approaches and methodology applied.

Traditionally, the society is assigned responsibility for solving health problems and diseases to the healthcare sector. Undoubtedly, the incorrect division of health care – not providing health care to those who need it most – is one of the key determinants of health. But the burden of disease, which is responsible for a significant number of premature deaths, arises largely as a result of the conditions which people live in. Conversely, the poor and unequal living conditions are the outcome of the inadequate social policies and programmes. Specific actions on the social determinants of health must involve all the levels of the government with all the spheres involved in it. The policies and programmes must cover all the key sectors of society, not just the health sector. The ministry of health and the relevant ministries are crucial for intracountry change. They are able to promote attitudes towards the social determinants of health at the highest level of society, can highlight the effectiveness of good practice, and can support the other ministries in developing policies that promote health equity. At the internatinal level, the World Health Organization serves as a global health institution and it lies at a global scale.

The limitations are present in a form of the access to the data mainly. This is the most considered issue in this point. If the data were complete with the longest history as possible, it would allow the deeper analytical processes in order to obtain the more exact outcome. The world is changing very fast and it is often unclear what impact social and economic impacts will have and policy changes on health in general and on health inequalities within the countries and also between the countries. Activities in the field of the social determinants of health will be more effective if the basic data systems are present, including the registration and monitoring of health and social inequalities determinants of health, more mechanisms are put in place in order to ensure comprehensibility and to apply data to create more effective policies, systems and programmes. Education on the social determinants of health is very important.

If it is required to follow whatever further change, we are ready to do it and we appreciated every your comment.

Reviewer 2 Report

I thank the authors for their engagement in this important research consideration.  

However, most of my comments and questions have not been addressed directly. I suggest that the authors reconsider all the questions and comments in my first review and try to address them more directly and appropriately. A case in point is the presentation in the manuscript of a separate section for introduction and literature review. I have attached an article from the International Committee of Medical Journal Editors (ICMJE), which is widely followed in public health publications in the West.

Please read page 14 of the article attached along with this review for your consideration. The website link  is

http://www.icmje.org/about-icmje/faqs/icmje-recommendations/

Of course, the authors are not obliged to follow ICMJE guidelines unless the publisher is a member of ICMJE.

Author Response

Dear reviewer
Thank you very much for your valuable comments. We appreciate your proposal and we have gone through all the recommendations of the International Committee of Medical Journal Editors. They represent a very good procedure for construction of the manuscript. We will keep it in our minds in the further work in this field.
Thank you very much for your suggestion. We hope that these revisions improve the paper so that you now deem it worthy of publication in International Journal of Environmental Research and Public Health and our revision has improved the paper to a level of your satisfaction. We have made the following changes in the text according to your suggestions:
1. We could join together these two sections. But we selected this way, because firstly, we would like to introduce the discussed topic to reader generally and subsequently, secondly, we would like to get reader into the theoretical background of the topic. Therefore, we have distinguished these two parts. If it was neccessary, we could join them. Another point is to make all the sections clear for reader and hence, we divided it into the two parts.
2. Social stratification determines the various approaches to healthcare that results in injustice in the prevention and treatment of the diseases, respectively. The health inequalities represent the result of the social norms, practices and policies that promote unfair distribution and access to wealth, power and social resources. The intensity of their impact and the types of these policies can enlarge the health inequalities. The socioeconomic determinants are the important indicators of these processes. Therefore, an examining of their effects on mortality make it possible to assess the extent of the health inequalities as well as the quality and efficiency of the health system of the country. These consistent facts represent also the motive for the analysis of this study, whose main aim is to examine the causal effects of the selected socioeconomic determinants of mortality in the European countries. An existence of the significant differences between the countries is investigated, so that it is possible to assess these differences in the government policies among the countries. It is an examination of the initial differences with an ambition to specify the factors that are most sensitive to change the mortality rate values which can bring many interesting findings for policymakers and it is important for the implementation of the outcome of the comparative analyses too. These ones will also be important in setting up processes to eliminate the health inequalities between the countries declared by the major international institutions.
3. It is quite difficult to answer this question. The socio-economic inequalities in health are still a topical issue and persist in the developed countries too. They represent one of the most serious and most unfair forms of social inequality and they bring great social costs at the same time. There is a close link between the health inequalities and the overall health status of the population – the greater the health inequality, the worse the health of the country's population as a whole. The presence of the socio-economic inequalities in health is one of the well-documented and repeatedly confirmed research hypotheses, despite the high variability in measurement approaches. Sociological research, which plays one of the most important roles in this area, rarely analyses the objective indicators of health. A more frequently used way is a subjective indication, which relates to the presence of the disease, health restrictions and the assessment of one's own health or the determination of the degree of satisfaction. Therefore, subjective health assessment belongs to the standardly used proxy indicators of objective health with confirmed strong validity and reliability. Many studies examining the influence of socio-economic determinants on subjective health assessments work with only simple indicator, whilst working with aggregated indicators represents still an exception. This is also due to methodological complexity, while the quality and completeness of international databases is also important.
4. Traditionally, the society is assigned responsibility for solving health problems and diseases to the healthcare sector. Undoubtedly, the incorrect division of health care – not providing health care to those who need it most – is one of the key determinants of health. But the burden of disease, which is responsible for a significant number of premature deaths, arises largely as a result of the conditions which people live in. Conversely, the poor and unequal living conditions are the outcome of the inadequate social policies and programmes. Specific actions on the social determinants of health must involve all the levels of the government with all the spheres involved in it. The policies and programmes must cover all the key sectors of society, not just the health sector. The ministry of health and the relevant ministries are crucial for intracountry change. They are able to promote attitudes towards the social determinants of health at the highest level of society, can highlight the effectiveness of good practice, and can support the other ministries in developing policies that promote health equity. At the internatinal level, the World Health Organization serves as a global health institution and it lies at a global scale. After completing the paper, we have found out that this is issue so complex, it would employ a very comprehensive study to cover all the examined field.
5. The research design is based on the common kind of the analysis of sensitivitiy in a form of the regression analysis. We decided that an application of the standard models would bring the interesting results altogether with the lagged regression models. They are specific in revealing of the delayed response in a relation to the explained variable.
6. The main aim of the paper is to create a basement for further research, because it is a very comprehensive topic, it is impossible to cover all the aspects of the examined dimensions. Therefore, the main research questions are rather general views on the explored variables. We have added the paragraph involving aiming of the analysis at the end of the Introduction section.
7. The independent variables are the values of the standardised mortality rate. We distinguish the overall mortality for the both sexes, the standardised mortality rate for the female sex, and the standardised mortality rate for the male sex. It is very important, because the both sexes have the different predispositions for the particular diagnoses. It is common approach absolutely.
8. The dependent variables, that is, the explanatory variables are the gross domestic product per capita, the total healthcare expenditures as a share of the gross domestic product, an unemployment rate, and the healthcare system financing model. They are listed in the Data and Methodology section.
9. Sampling unit is defined according to the explained variable, that is, the standardised mortality rate – a number of deaths per 100,000 inhabitants according to the standard European population designed by Eurostat. It is very important to keep this in order to be compared with the other studies. The employed sampling method is the standard methodology in a form of the sensitivity analysis. The sample size is defined according to the accessible data through the common public database. We have covered the time period beginning in 1992 and ending in 2015.
10. The R software environment was employed in order to carry out the analysis. It is very good statistical software to perform whatever analysis.
11. This question is very complex and to answer it, it requires a very comprehensive reply. It depends on an angle of view, because each aspect can bring up little different answer with alternation of the independent social determinants. It is very important to look at the models not only separately, but also comparably and altogether. Some of the social determinants are causal and interconnected each other and it is required to understand them in this way. Experience shows that countries that do not have fundamental data on mortality and morbidity stratified by the socio-economic indicators are possessing difficulties in a field of their health status challenges. The countries with the most serious health problems lack quality data. Many countries do not even have a basic system of birth and death records. Defective birth registration systems have a significant impact on the health and development of children.
12. We have not applied the four decades old data because it would be quite uninterpretable. But the longer examined period is, the more acceptable it is. The twenty-three-year time span can serve as a good basement for the exploration of the influences of the particular explanatory variables.
13. Yes, this could be point, but we have tried to find the similar studies and we would be very glad if our study can serve as an example for further research in other territories or for similar socioeconomic research. Knowledge of the state of health from a global, regional, national and local perspective, the programmes applicable in the particular situation and how it effectively affects health inequalities through the social determinants of health is a key area, which the contemporary research focuses on. It also serves as a support for all the recommendations for the policymakers. This research is perhaps the most needed activity in this field. But more than so-called academic exercise, research is required in order to bring new knowledge and can be disseminated in all accessible ways among the above-mentioned stakeholders. Research and knowledge of the social determinants of health and ways to achieve health equity should be based on ongoing discussion between academics and people from practice, but also on new methodologies – recognition and use of a wide range of knowledge, sex inequality in research practices also role here. Activities in this area include also building and disseminating knowledge about the social determinants of health. Research grants should be awarded to the projects on the social determinants of health, support global health observatories and multilateral, national and local intersectoral projects aimed at developing and validating indicators of social determinants of health and evaluating the impact of interventions. Also they should be focused at establishment and development of virtual networks and associations organised on the principle of open access that would be managed in such a way as to allow access for all income groups. This should contribute to halt the brain drain from the low-income and middle-income countries and to address the issue of sex inequalities and their eliminatation.
14. The limitations are present in a form of the access to the data mainly. This is the most considered issue in this point. If the data were complete with the longest history as possible, it would allow the deeper analytical processes in order to obtain the more exact outcome. The world is changing very fast and it is often unclear what impact social and economic impacts will have and policy changes on health in general and on health inequalities within the countries and also between the countries. Activities in the field of the social determinants of health will be more effective if the basic data systems are present, including the registration and monitoring of health and social inequalities determinants of health, more mechanisms are put in place in order to ensure comprehensibility and to apply data to create more effective policies, systems and programmes. Education on the social determinants of health is very important.

If it is required to follow whatever further change, we are ready to do it and we appreciated every your comment.

Round 3

Reviewer 2 Report

There is no need to present the introduction and literature review separately unless this is required by the publisher.  Both can be combined under Introduction/Background information.